# A systematic review comparing the performance of alternative blackfly (*Simulium*) trapping methods against the standard human landing catch (HLC) for onchocerciasis surveillance

Irene Kyomuhangi[1,2], Amro Mustafa[3], Frances M. Hawkes[4]*

1 Centre for Health Informatics, Computing, and Statistics (CHICAS), Lancaster University, Lancaster, United Kingdom, 2 Bennett Institute for Applied Data Science, University of Oxford, Oxford, United Kingdom, 3 University of Bath, Bath, United Kingdom, 4 Natural Resources Institute, University of Greenwich, Greenwich, United Kingdom

* f.m.hawkes@greenwich.ac.uk

## Abstract

Despite decades of control efforts, onchocerciasis remains a major public health concern in Africa, the Americas, and Yemen. Human Landing Catch (HLC) is the primary method for collecting blackflies and is central to surveillance. However, HLC raises ethical concerns due to collectors' exposure to painful and potentially infectious bites and faces operational challenges in areas of very low or high transmission. Consequently, several alternative blackfly trapping methods have been investigated, but no comprehensive synthesis comparing their effectiveness against standard HLC across studies has been conducted. Therefore, we performed a systematic review comparing the performance of alternative blackfly traps with standard HLC. A systematic review (PROSPERO registration number: CRD420261294895) of literature published in Scopus, PubMed, and Web of Science up to December 2025 was supplemented by an expert-provided reference list. From 166 records, 62 were screened, and 13 studies (comprising 79 comparisons with standard HLC) met inclusion criteria. Alternative traps included light traps, Bellec traps, tent traps baited with humans or cows, Esperanza Window Traps, Host Decoy Traps, electric nets, and modified HLC. Most comparisons (75.9%) found alternative traps to be less effective than standard HLC, with statistical analyses often supporting these differences, although nearly half lacked formal significance testing. Variation in study design—including trap placement, rotation, and trapping duration—and inconsistent reporting of key variables such as season, habitat, and species limited direct comparisons. Some studies indicated that increasing trap density or deployment duration of Esperanza Window Traps could improve effectiveness. While HLC remains the most effective method, its ethical and operational limitations highlight the need for reliable alternatives. Most existing traps underperform relative to HLC, but modifications based on deeper understanding of blackfly behaviour and ecology could improve performance.

**Data availability statement:** The dataset containing extracted data from the 13 studies included in this systematic review is available on Zenodo. DOI: https://doi.org/10.5281/zenodo.18631035.

**Funding:** This work was supported by the Gates Foundation grant reference INV-037397 to IK and FMH. Health Data Research UK provided support for AM under the Health Data Science Black Internship Programme. The funders had no role in study design, data collection and analysis, decision to publish, or preparation of the manuscript.

**Competing interests:** The authors have declared that no competing interests exist.

Future research should focus on standardizing trap evaluation methods, exploring species-specific behaviours, and assessing scalability to develop ethical, scalable tools for onchocerciasis surveillance.

## Author summary

After decades of interventions against onchocerciasis, also called 'river blindness', many countries hope to demonstrate they have interrupted disease transmission, allowing them to stop delivering interventions and divert healthcare resources elsewhere. To verify if this has been achieved, the World Health Organization requires evidence that the causative parasite, *Onchocerca volvulus*, is no longer being transmitted by the insect vector, blackflies of the genus *Simulium*. This necessitates collecting large numbers of blackflies, which is usually achieved using a manual method called the Human Landing Catch (HLC), using a human as both bait and collector. Several alternative blackfly trapping methods have been developed to avoid the ethical and operational issues associated with exposing a person to potentially infective bites. We systematically reviewed the literature to compare the effectiveness of these alternative traps relative to the standard HLC. We found there to be considerable variability in the design of studies to test these, which made study-to-study comparisons difficult, and suggest standardized study designs would allow future comparisons and meta-analysis. Although the majority of published data suggests that the HLC is the most effective method, there is some evidence that using multiple traps could match HLC results, and trap design could be improved by including additional features to which blackflies are attracted.

## Introduction

Onchocerciasis, or "river blindness," is a parasitic disease caused by the filarial worm *Onchocerca volvulus*, spread by blackflies of the *Simulium* genus and can lead to blindness. Decades of interventions, such as mass drug administration (MDA) with microfilaricidal drugs and vector control to control blackflies, have substantially decreased the burden of onchocerciasis in the Americas and Africa. Nevertheless, the disease continues to pose a major public health challenge, as over 250 million people still require preventive treatment annually to interrupt its transmission [1].

The World Health Organization (WHO) has set an objective to eliminate onchocerciasis, with MDA being a key strategy to achieve this [2]. In the context of disease elimination, WHO guidelines for stopping MDA rely on entomological and serological monitoring. Specifically, for entomological surveillance, the guidelines for confirming elimination require collecting at least 6,000 blackflies from a transmission zone, which must then be tested and verified to be free of infective *O. volvulus* larvae [3].

This surveillance, which includes monitoring the prevalence of infective *O. volvulus* in blackflies, blackfly biting rates, and other entomological indicators, is crucial to onchocerciasis control programmes [4].

The conventional approach for collecting blackflies is the Human Landing Catch (HLC), in which trained human collectors act as bait by exposing their legs to attract blackflies, which they then capture. However, this method has some limitations. In areas with low transmission or seasonal blackfly activity, collectors may capture very few blackflies and struggle to reach the target of 6,000 [4,5]. Conversely, when blackfly densities are extremely high, it is unknown whether blackfly catches accurately represent the actual biting density. Importantly, the HLC method raises ethical issues regarding the safety of the collectors [6,7]. Despite receiving microfilaricide treatment and being trained to capture blackflies before they bite, the collectors still endure bites from blackflies and other anthropophilic haematophagous insects, possibly including other vector species. Anecdotal observations indicate a small but consistent proportion of collected blackflies contain bloodmeals that are likely to originate from the collector. Blackfly bites can be quite painful [8], and the collectors are subjected to this exposure for extended periods, which may lead to pruritus and other symptoms [9]. For these reasons, some onchocerciasis-endemic countries will no longer grant ethical approval for using the HLC method to collect blackflies (M. Imhansoloeva, pers. comm.).

Over the past four decades, various alternative methods for capturing blackflies have been trialled and implemented with differing levels of success [4] (Fig 1). These include Light Traps, which exploit potential phototactic behaviour of gravid females and newly emerged flies [10]; Bellec Traps, consisting of an aluminium panel coated with glue and placed near rivers, where gravid female blackflies seeking oviposition sites become trapped [11–13]; tent traps that use humans or cows as bait to attract host-seeking blackflies [6,13,14]; Esperanza Window Traps (EWTs), which use visual cues and olfactory attractants to lure blackflies onto sticky surfaces [15–17]; the Host Decoy Trap (HDT), which uses a human attractant to draw host-seeking flies that then land on a heated adhesive surface and are captured [18]; electric nets, which combine a human or visual attractant with electric grids to lure and kill blackflies [14,19]; and modified Human Landing Catch (HLC), designed to protect collectors from bites by having them wear trousers without their legs exposed or apply oil on their legs as a deterrent [20,21].

Individual studies have evaluated the performance of alternative blackfly trapping methods, and in some cases compared their performance against the standard HLC. However, to our knowledge, no existing research has synthesized this information to offer a comprehensive review of how these methods perform relative to the standard HLC. Therefore, this systematic review aims to evaluate the performance of alternative blackfly trapping methods in comparison to the standard HLC across different epidemiological contexts.

## Methods

This systematic review was registered with PROSPERO (registration number: CRD420261294895) [23] on 27 January 2026. The preferred reporting items for systematic reviews are provided (S1 Table).

### Literature search

We conducted a literature search across Scopus, PubMed, and Web of Science databases for related articles published up to 17 December 2025. The search used keywords related to blackfly trapping methods (Table 1). The literature search identified 160 records between all three databases (Fig 2).

### Expert list

On 03 July 2024, a list of 12 references - including published and unpublished work - was provided by the senior entomologist on the study (FH). This list was based on expert knowledge of significant work related to the research question. Of the 12 references, 6 did not overlap with the literature search and so were assessed as additional literature alongside the 160 records identified in the literature search (Fig 2) and were entered into the subsequent screening and review process.

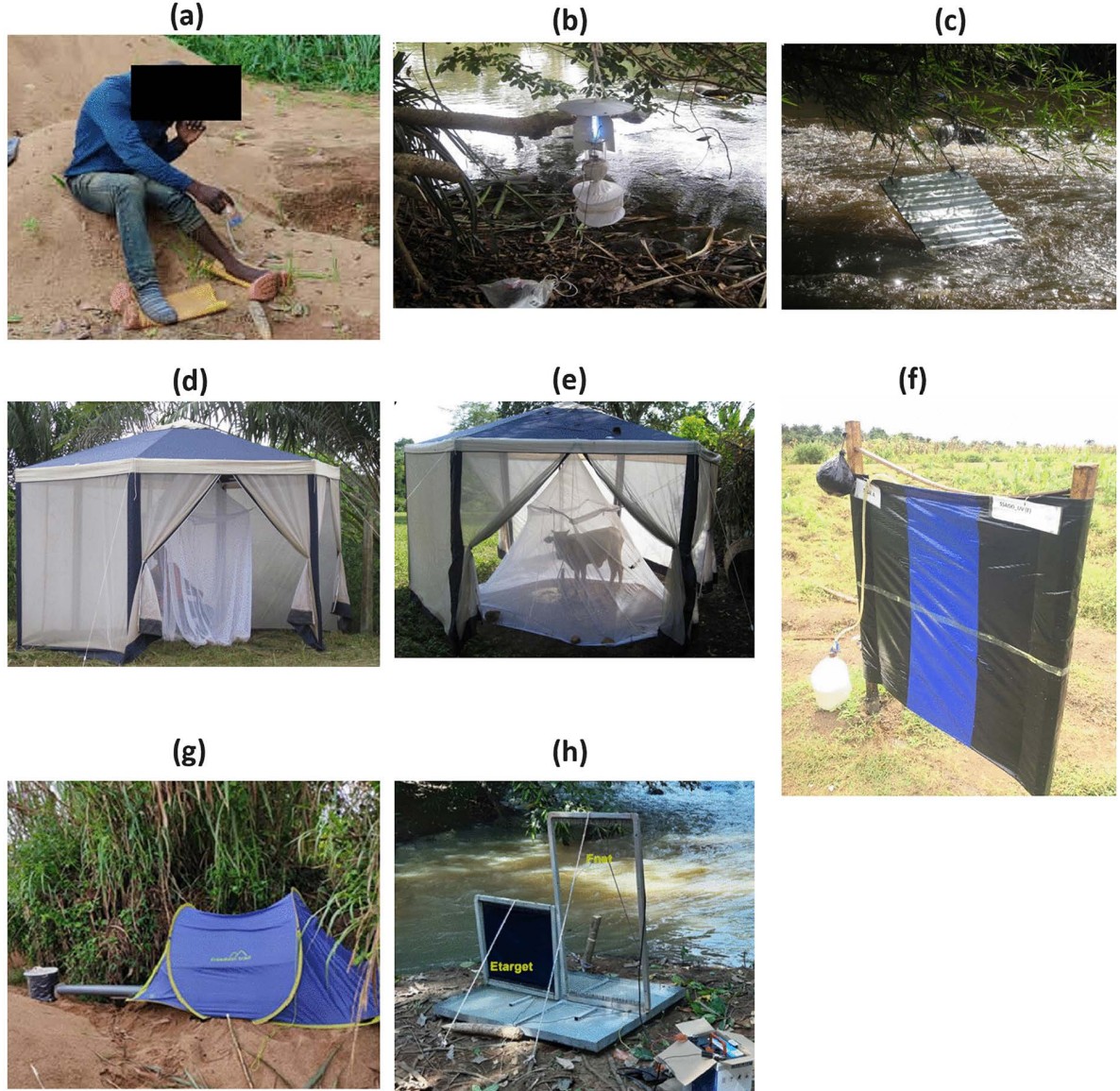

**Fig 1. Different blackfly trapping methods. (a)** the standard Human Landing Catch [18]; **(b)** Monk's Wood (light) trap [13]; **(c)** Bellec trap situated above rapids [13]; **(d)** human-baited tent [13]; **(e)** cow-baited tent [13]; **(f)** Esperanza Window Trap [22]; **(g)** Host Decoy Trap [18]; **(h)** electric net [19]. The images have been reproduced in accordance with their respective licences (see Acknowledgements).

**Table 1. Search terms used across Scopus, PubMed and Web of Science databases.**

| simulium OR simuliidae OR blackfl* OR black flies OR black fly | AND | trap OR capture OR HLC OR human landing catch OR landing collection* OR vector collector | AND | oncho* OR river blindness |
|---|---|---|---|---|

PLOS Neglected Tropical Diseases

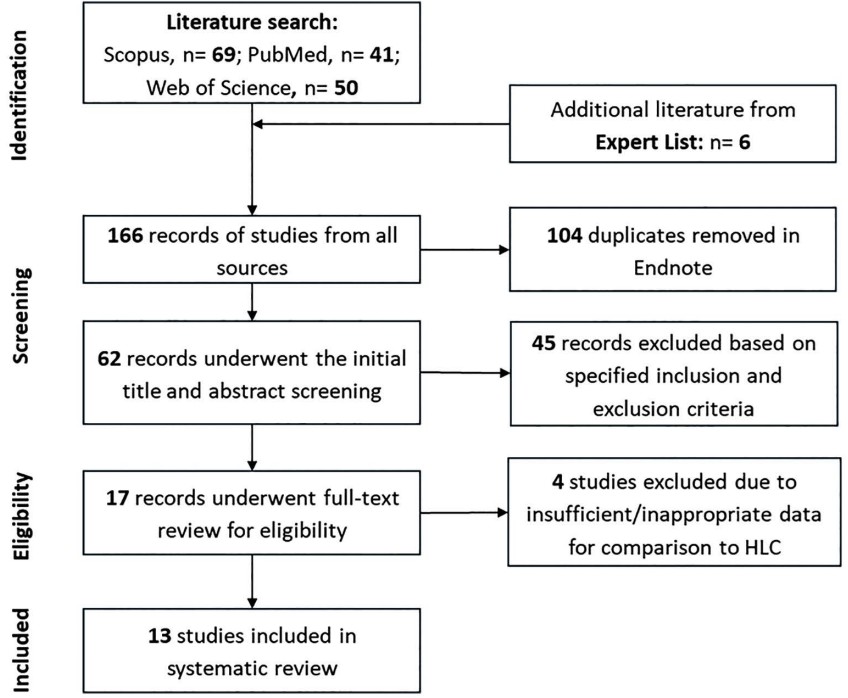

**Fig 2. A flowchart showing the selection of studies to include in the systematic review.**

## Screening and review

Endnote 21.4 [24] was used for reference management, including removing duplicate references. A total of 62 records from both the literature search and the expert list underwent an initial title and abstract screening to determine their relevance based on inclusion and exclusion criteria specified in Table 2. This initial screening was performed independently by two investigators (IK and AM) using Rayyan [25], and any discrepancies were addressed through discussions among AM, IK, and FH. Full text review, and data extraction using Microsoft Excel, were conducted on 13 studies by IK.

## Data extraction

Our aim was to review studies that made direct comparisons between HLC and alternative trap types. One reviewer (IK) extracted data using a custom Microsoft Excel form. We define a 'comparison' as an instance when the performance of

**Table 2. Inclusion and exclusion criteria applied to studies from the literature search.**

| Inclusion | Exclusion |
|---|---|
| Empirical studies containing data on blackfly catches | Non-empirical studies (e.g., reviews) |
| Blackfly trapping is conducted in field settings | Blackfly trapping is conducted only in a laboratory setting |
| Contains comparison between the standard HLC and one or more alternative trapping methods in the same experiment* | Contains data on HLC only, or contains data on alternative trapping methods but excludes HLC, or data on trapping methods is from different experiments* |

*Same experiment is taken to mean that data collection is conducted at the same location and during the same collection period (timeframe).

an alternative trapping method was compared against the standard HLC within the same experiment. A study could have multiple experiments (e.g., testing different versions of EWTs, which were then individually compared against the standard HLC). In this case, each experiment constituted its own comparison. For experiments where multiple metrics were compared to the standard HLC (e.g., if authors compared total catch and mean hourly catch from the same experiment), this was taken to be the same comparison. Extracted variables included study design; location; habitat; year and season of data collection; trapping methods; trap rotation and proximity; number of traps and HLC collectors; species identification; outcomes measured; statistical methods used; and reported results of comparisons between alternative traps and the standard HLC. The primary outcome was the blackfly catch metric reported by each study (total catch, mean hourly catch, mean daily catch, or monthly biting rate). Where multiple compatible measures were reported, all relevant results were extracted.

### Synthesis

Due to substantial heterogeneity in trap designs, outcome measures, sampling schemes, and incomplete reporting, quantitative meta-analysis was not appropriate. We therefore performed a structured narrative synthesis: tabulation of study characteristics and outcomes, and calculation of descriptive proportions. Extracted metrics were used as reported, no conversions were performed. Missing or unclear data were recorded as such (authors were not contacted to provide missing data, consistent with the PROSPERO plan [23]); no imputation was performed.

### Risk of bias approach

A formal risk-of-bias assessment of individual studies was not conducted. This decision was prespecified in the PROSPERO plan [23] because the primary aim of the review was descriptive synthesis rather than quantitative meta-analysis, and the included studies were highly heterogeneous in design and outcome reporting. Instead, methodological limitations of included studies are described narratively in the Results and Discussion.

### Reporting bias

Formal assessment of publication or reporting bias was not feasible because of the small number of studies and heterogeneity of outcome measures.

### Certainty of evidence

We did not perform a formal GRADE assessment because of heterogeneity in study designs and outcomes. The narrative synthesis and the Discussion summarise factors that reduce confidence in the evidence.

## Results

### Characteristics of the studies and comparisons included in the systematic review

The 13 studies selected for the systematic review were conducted in the Americas and Africa (Table 3), with data collection conducted between 2009 and 2023, and publication of results occurring between 2013 and 2024. Each study provided a comparison between at least 1 alternative blackfly trapping method and the standard HLC. Results from 79 comparisons between a trapping method and HLC were extracted from the 13 studies (Table 3).

Most comparisons between trapping methods were conducted in savannah habitats (49.4%), and during the dry season (51.9%) [Table 4].

Across the studies, there was variability in experimental design. For example, the distance between the alternative trapping methods and the standard human landing catch (HLC) differed, with reported distances ranging from 2 meters to 100 meters. Additionally, in 6 out of the 13 studies included in the systematic review, the distance between the alternative trapping method and HLC was not specified [6,14,16,17,19,27].

**Table 3. Characteristics of the studies included in the systematic review.**

| Region | Country | Number of studies | Number of unique study sites | Number of comparisons at the study sites | Reference |
|---|---|---|---|---|---|
| Americas | Brazil | 1 | 2 | 2 | Nascimento-Carvalho et al, 2017 [6] |
| | Mexico | 3 | 2 | 7 | Rodríguez-Pérez et al, 2013 [15]; Rodríguez-Pérez et al, 2014 [5]; Rodríguez-Pérez et al, 2017 [26] |
| Africa | Burkina Faso | 2 | 2 | 16 | Toé et al, 2014 [16]; Koala et al, 2022 [19] |
| | Cameroon | 2 | 2 | 8 | Talom et al, 2021 [18]; Atekem et al, 2024 [20] |
| | Ghana | 1 | 7 | 22 | Lamberton et al, 2014 [14] |
| | Nigeria | 1 | 4 | 5 | Adeleke et al, 2024 [22] |
| | Tanzania | 1* | 1 | 1 | Hendy et al, 2017 [17] |
| | Uganda | 3* | 7 | 18 | Hendy et al, 2017 [17]; Loum et al, 2017 [28]; Loum et al, 2019 [27] |

*One study by Hendy et al [17] contains data from both Tanzania and Uganda.

**Table 4. Habitats and seasons in which the comparisons between trapping methods and HLC were conducted in the studies included in the systematic review.**

| Region | Country | Number of comparisons conducted in different habitats | | | | Number of comparisons conducted in different seasons | | | |
|---|---|---|---|---|---|---|---|---|---|
| | | forested | savannah | other* | unspecified | wet | dry | mixed | unspecified |
| Americas | Brazil | 1 | | 1 | | | | | 2 |
| | Mexico | 2 | | | 5 | | 7 | | |
| Africa | Burkina Faso | | 16 | | | | 12 | 4 | |
| | Cameroon | 2 | | | 6 | 3 | 3 | | 2 |
| | Ghana | 10 | 4 | 4 | 4 | 4 | 18 | | |
| | Nigeria | 2 | 1 | 2 | | | | | 5 |
| | Tanzania | 1 | | | | | 1 | | |
| | Uganda | | 18 | | | 10 | | 6 | 2 |
| **Total** | **8** | **18** | **39** | **7** | **15** | **17** | **41** | **10** | **11** |

*other habitats were mainly transition zones or areas with recent changes to the landscape (e.g., due to agricultural practices).

Furthermore, the duration of blackfly trapping differed across studies, spanning from as brief as 3 days to as long as 60 days. Some collections were conducted on consecutive days, while others involved intervals of up to a week between collection days.

Rotation of trapping methods was reported in 97.4% of the comparisons. However, the approach to rotation varied. In some cases, only the alternative trapping methods being compared were rotated, while the standard HLC remained constant (or vice versa) [6,14,16,26–28], whereas other studies rotated all trapping methods, including the standard HLC [5,15,17–20,22].

The identification of *blackfly* species differed among studies and comparisons. The majority of authors specified particular species or species complexes, such as *S. ochraceum s.l.*, *S. damnosum s.l.*, *S. damnosum s.s.*, *S. bovis*, *S. sirbanum*, and *S. squamosum,* whereas others referred more generally to the genus or family level, using terms like Simuliidae or simply "*Simulium*" [6,20].

## Alternative trapping methods compared against the standard HLC

Among the 79 comparisons in the systematic review, the EWT was the dominant trapping method compared against the standard HLC (45.6%); however, there was substantial variation in the versions used, including differences in colour, size (of the trap and the stripes on the trap), the type of bait, and the addition of carbon dioxide (Table 5).

Other types of trapping methods that were compared to the HLC include the MosqTent trap [6], the host decoy trap (HDT) [18], electric grid E-target traps [19], a modified version of the HLC where collectors wore blue and black trousers [20], as well as tent traps baited with humans or cows [14].

## Comparing outcomes between the alternative trapping method vs standard HLC

The outcomes used for the comparisons also varied, and included total catches, mean hourly catch, mean daily catch, and monthly biting rate (Table 6). Total catch was the most common outcome among the 79 comparisons (43.0%).

In most comparisons (75.9%), blackfly collections from a single alternative trapping method were evaluated against collections from the standard HLC in a 1:1 ratio in the number of traps to HLCs [15,16,18–20,22,28]. It is important to note that even when two HLC collectors took turns conducting the collection (following the standard protocol), this still counted as a single HLC collection. In contrast, 22.7% of comparisons used a 2:1 ratio, with two alternative traps compared to 1 HLC collection [17,22].

In the 79 comparisons included in the systematic review, determination of statistically significant differences between outcomes from the alternative trapping method Vs the standard HLC included the use of t-tests [15,16], Mann-Whitney test [6], Kolmogorov-Smirnov test [6], ANOVA, Tukey's multiple comparison test, and negative binomial models [5,17–20,28]. Across the comparisons, there was variation in whether formal statistical comparison had been conducted or reported. Significance testing to assess differences between outcomes was not performed or reported in 49.4% of the comparisons [14,26,27]. Where statistical comparison was conducted and reported, this is indicated in Table 7.

## Findings of the comparisons

Among the 79 comparisons, the majority (75.9%) indicated that the trapping method was less effective in capturing black-flies when compared to the standard HLC method. Specifically, 36 comparisons indicated that the trapping method was less effective (statistical significance not reported), whereas another 24 comparisons showed that the trapping method was significantly less effective than the HLC method (statistical significance reported) [Table 7].

**Table 5. Different versions of the EWT in the studies included in the systematic review.**

| Variable | Versions of the variable |
|---|---|
| colour | the pattern of blue and/or black |
| size | size of the trap |
| | size and shape of the stripes on the trap |
| bait | no lure |
| | BG lure |
| | worn socks |
| | aroma beads |
| | tangle trap |
| $CO_2$ | yeast generated |
| | $CO_2$ mimic |
| | no $CO_2$ |

**Table 6. Outcomes used for the 79 comparisons in the systematic review. Full extraction data are available on Zenodo (see link in Data and software availability).**

| Study number | Reference | Trap types in study | Number of comparisons in study | Outcome of comparison | | | |
|---|---|---|---|---|---|---|---|
| | | | | Mean hourly catch | Mean daily catch | Mean monthly biting rate | Total catch |
| 1 | Rodríguez-Pérez et al, 2017 [26] | EWTs | 4 | | | | 4 |
| 2 | Loum et al, 2019 [27] | EWTs | 2 | 2 | | | |
| 3 | Talom et al, 2021 [18] | EWTs and HDTs | 2 | | 2 | | |
| 4 | Koala et al, 2022 [19] | Electric grid Etarget traps | 12 | | 6 | | 6 |
| 5 | Adeleke et al, 2024 [22] | EWTs | 5 | | 5 | | |
| 6 | Atekem et al, 2024 [20] | Modified HLC where collectors wear blue and black trousers | 6 | | | | 6 |
| 7 | Rodríguez-Pérez et al, 2013 [15] | EWTs | 2 | 2 | | | |
| 8 | Toé et al, 2014 [16] | EWTs | 4 | | 4 | | |
| 9 | Hendy et al, 2017 [17] | EWTs | 11 | | | | 11 |
| 10 | Loum et al, 2017 [28] | EWTs | 6 | | | | 6 |
| 11 | Nascimento-Carvalho et al, 2017 [6] | MosqTent | 2 | | 2 | | |
| 12 | Rodríguez-Pérez et al, 2014 [5] | EWTs | 1 | | | | 1 |
| 13 | Lamberton et al, 2014 [14] | Tent traps baited with humans or cows | 22 | | | 22 | |
| **Total** | | | **79** | **4** | **19** | **22** | **34** |

**Table 7. Findings from comparison of alternative trapping methods to the standard HLC method.**

| Trap type | Number of comparisons | Performance of trap vs HLC | | | | | |
|---|---|---|---|---|---|---|---|
| | | Less effective^ | | No significant† difference | More effective ß | Significantly† more effective | Mixed results* |
| | | Less effective ß | Significantly† less effective | | | | |
| EWT | 36 | 14 | 11 | 8 | 2 | 1 | |
| MosqTent | 2 | | | 1 | | | 1 |
| HDT | 1 | | | 1 | | | |
| Electric grid Etarget traps | 12 | | 9 | 3 | | | |
| Modified HLC where collectors wear blue and black trousers | 6 | | 4 | 2 | | | |
| Human tent trap | 11 | 11 | | | | | |
| Cow tent trap | 11 | 11 | | | | | |
| **Total** | **79** | **36** | **24** | **15** | **2** | **1** | **1** |

^ effective in the number of flies caught (reported as total catch, or mean hourly/daily/monthly catch).

ß no statistical significance reported.

† a statistical significance reported.

*In one paper [6], the MosqTent trap was significantly better than HLC when mean daily catch was compared, but not significantly different when total catch over the entire study period was compared.

Due to the variability in study site characteristics like habitat and season, the outcomes used in the comparisons, how the blackfly collections were conducted, as well as how comparisons were conducted and reported, it was difficult to discern which factors were associated with comparable or improved trap performance relative to the standard HLC, where these outcomes were indicated in Table 7.

## Discussion

This systematic review synthesizes data from 79 comparisons across 13 published studies, evaluating the performance of different alternative blackfly trapping methods against the standard HLC. The primary finding is that, across a range of epidemiological settings and habitats, most alternative traps, given their current designs and deployment methods, are generally less effective than the standard HLC in capturing blackflies. Consequently, they do not yet match the sampling efficiency of the standard HLC, which remains the standard method for blackfly collection. Nevertheless, this review also elucidates important nuances, limitations, and potential avenues for future research aimed at improving alternative blackfly trapping methods.

The heterogeneity observed across studies complicates direct comparisons and precludes robust meta-analyses. Specifically, variations in experimental design, including differences in the distance between trapping methods, duration of trapping, and the number of traps used, all affect trap performance metrics. These variations make it difficult to determine whether observed differences in performance are attributable to the traps themselves or to different aspects of the experimental design. Future research should prioritize standardised methodologies, including consistent trap placement and rotation protocols. Such standardisation would facilitate more direct comparison across studies and help inform evidence-based guidelines for blackfly surveillance in different endemic settings.

Inconsistency in reporting key variables such as habitat type, seasonality, and species identification limits the interpretability and generalizability of findings. Many studies did not specify the distance between traps, which is important because trap performance may be affected by their proximity to one another, as is the case for tsetse fly (Glossinidae) [29] and mosquito (Culicidae) [30] traps. Differences in habitat types (forested vs. savannah), seasons (wet vs. dry), and local species compositions are reported to influence trap catches in blackflies and other vectors [31–33]. There was insufficient data to determine whether any of these specific factors have a consistent effect on trap performance relative to HLC. Further to the need for additional data, the lack of consistency in reporting highlights the need for harmonised reporting conventions to enable more meaningful cross-study comparisons.

Some studies also identified additional potential directions for future research. Notably, Loum et al, 2017 [27] reported that the combined blackfly catches from 3 EWTs yielded over 4 times more blackflies compared to 1 standard HLC over a 5-month collection period in Northern Uganda, indicating potential scalability advantages. At the same study site 2 years later, Loum et al, 2019 [27] replicated this finding, showing that 5 EWTs collected between 4–26 times the mean daily catch compared to 1 HLC. Similarly, Adeleke et al, 2024 [22] found that deploying 2 EWTs yielded similar catches to 1 standard HLC over a 9-day sampling period in Nigeria. These findings imply that, while individual EWT traps may be less effective, increasing trap density and/or collection times could improve their utility. Additional research is required to determine the extent to which this applies and whether it is relevant to other trap types.

Furthermore, a deeper understanding of blackfly behaviour and ecology is essential for improving trap design and deployment. Understanding the biological mechanisms driving vector behaviour, such as responses to host-seeking cues including visual stimuli and olfactory attractants, is essential for targeted optimisation of vector surveillance and control tools [34]. For instance, the variation in EWT versions (differing in colour, size, bait, and $CO_2$ sources) suggests that trap attractiveness can be fine-tuned, but systematic studies are needed to identify the most effective combinations [35]. Additionally, the species composition at the study sites likely impacts trap effectiveness. Gaining a deeper understanding of the fundamental behaviours of individual species could guide the development and implementation of alternative, context-specific trapping methods that may be more effective in the appropriate setting.

While the standard HLC currently remains the most effective method for blackfly collection, its limitations, including safety risks to collectors and logistical challenges, highlight the importance of developing reliable alternative traps. Research currently in progress on this topic will add to the body of data available in the public domain in due course. In parallel, cost-effectiveness analyses, considering trap durability, maintenance, and operational feasibility, are also needed to assess their suitability for large-scale surveillance, as are accurate risk profiles for vector collectors, regardless of the collection method in use. The latter may help to inform additional protective measures for collectors until alternative traps are more widely adopted. Guidance from ethical review boards on their expectations on the use-cases for and possible phase-out of HLCs would enable researchers and national onchocerciasis programmes to prioritise operational research to ready alternatives for widespread use.

Lastly, this review has some limitations. Although we searched three major bibliographic databases and supplemented these with an expert-provided reference list, we did not conduct a comprehensive search of all grey literature sources, and some unpublished field evaluations may have been identified. Data extraction was primarily conducted by a single reviewer, which may introduce the potential for error despite efforts to ensure consistency. In line with our prespecified protocol, we did not perform a formal GRADE assessment of certainty. These factors should be considered when inter-preting the findings, and they reinforce the need for more standardised, transparently reported evaluations of alternative blackfly trapping methods to strengthen the evidence base for surveillance decision-making.

## Conclusion

Overall, this systematic review indicates that while current alternative blackfly trapping methods are generally less effective than the standard HLC, strategic modifications, backed by a more robust evidence base for the biological basis of the trap design, as well as optimizing trap density could improve their utility. Future meta-analyses and programmatic confidence to adopt new tools will be contingent on additional data and more standardised protocols being adopted in trap evaluation studies. Developing and validating more effective, ethical, and scalable trapping methods will be critical for enhancing entomological surveillance, verifying elimination thresholds, and sustaining onchocerciasis control and elimina-tion gains. Continued research integrating behavioural ecology, technological innovation, and operational feasibility will be essential in achieving these goals.

## Supporting information

**S1 Table. PRISMA Checklist.** The PRISMA Checklist is licensed under a CCBY 4.0 license (https://creativecommons.org/licenses/by/4.0/deed.en) and attributed to Page MJ, McKenzie JE, Bossuyt PM, Boutron I, Hoffmann TC, Mulrow CD, et al. The PRISMA 2020 statement: an updated guideline for reporting systematic reviews. *BMJ* 2021;372:n71, https://doi.org/10.1136/bmj.n71.
(DOCX)

## Acknowledgments

The images in Fig 1b, 1c, 1d, and 1e are derived from Lamberton, et al., 2015 [12], licensed under the terms of the Creative Commons Attribution License https://creativecommons.org/licenses/by/4.0/. The images were not modified.

The images in Fig 1a and 1g are derived from Talom, et al., 2021 [17] licensed under the terms of the Creative Commons Attribution License:https://creativecommons.org/licenses/by/4.0/. Fig 1 was modified to obscure the face of the HLC collector for privacy purposes, whereas Fig 1g remains unmodified.

The image in Fig 1h is derived from Koala, et al., 2022 [18] licensed under the terms of the Creative Commons Attribution License: https://creativecommons.org/licenses/by/4.0/. The image was not modified.

The image in Fig 1f is derived from Adeleke, et al., 2024 [21] licensed under the terms of the Creative Commons Attribution License: https://creativecommons.org/licenses/by/4.0/. The image was not modified.

# Author contributions

**Conceptualization:** Irene Kyomuhangi, Frances Hawkes.

**Data curation:** Irene Kyomuhangi.

**Formal analysis:** Irene Kyomuhangi.

**Funding acquisition:** Frances Hawkes.

**Investigation:** Irene Kyomuhangi, Amro Mustafa, Frances Hawkes.

**Methodology:** Irene Kyomuhangi, Amro Mustafa, Frances Hawkes.

**Project administration:** Irene Kyomuhangi, Frances Hawkes.

**Software:** Irene Kyomuhangi, Amro Mustafa.

**Supervision:** Irene Kyomuhangi, Frances Hawkes.

**Validation:** Irene Kyomuhangi, Frances Hawkes.

**Visualization:** Irene Kyomuhangi.

**Writing – original draft:** Irene Kyomuhangi.

**Writing – review & editing:** Amro Mustafa, Frances Hawkes.

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
