## [Decision Letter · Decision Letter 0]

9 Apr 2026

PNTD-D-26-00444

A systematic review comparing the performance of alternative blackfly (*Simulium*) trapping methods against the standard human landing catch (HLC) for onchocerciasis surveillance

Dear Dr. Hawkes,

Thank you for submitting your manuscript to PLOS Neglected Tropical Diseases. After careful consideration, we feel that it has merit but does not fully meet PLOS Neglected Tropical Diseases's publication criteria as it currently stands. Therefore, we invite you to submit a revised version of the manuscript that addresses the points raised during the review process.

Please submit your revised manuscript within by May 8. If you will need more time than this to complete your revisions, please reply to this message or contact the journal office at plosntds@plos.org. Please include the following items when submitting your revised manuscript:

We look forward to receiving your revised manuscript.

Kind regards,

Emeka John Dingwoke, Ph.D

Academic Editor

Amy Morrison

Section Editor

Shaden Kamhawi

co-Editor-in-Chief

Paul Brindley

co-Editor-in-Chief

**Additional Editor Comments:**

Dear Dr. Hawkes,

Thank you for submitting your manuscript. This is a timely and relevant study addressing an important gap in onchocerciasis surveillance, with clear objectives, appropriate design, and a well-structured synthesis. The manuscript is generally well written and provides useful insight into current evidence and existing knowledge gaps.

However, several issues require revision before the manuscript can be considered for acceptance. Greater clarity is needed regarding the literature search and how its comprehensiveness was ensured, particularly given that some relevant studies were identified outside the main search. Key methodological elements also require clearer definition, including what constitutes a “comparison,” how equivalence in sampling effort was determined, and how outcome measures were standardised across studies.

The presentation of results should be strengthened by explicitly linking interpretations to the individual studies. Where variability in study design, trap performance, or analytical approaches is discussed, the corresponding studies should be clearly cited, including within tables. Tables should be revised to include more detailed study characteristics and improve clarity and consistency. In addition, clarify whether any quantitative synthesis was explored for comparable trap types, or provide a brief justification if not undertaken.

Further revisions are required to improve conceptual accuracy and presentation. These include refining the use of “xenomonitoring” in the appropriate context, clarifying statements related to ivermectin, ensuring all key claims are adequately referenced, and correcting minor inconsistencies in text, figures, and geographic classifications. Addressing these points will strengthen the manuscript and improve its clarity and rigor.

**Journal Requirements:**

At this stage, the following Authors/Authors require contributions: Irene Kyomuhangi, Amro Mustafa, and Frances Hawkes. Please ensure that the full contributions of each author are acknowledged in the "Add/Edit/Remove Authors" section of our submission form.

4) We notice that your supplementary Tables are included in the manuscript file. Please remove them and upload them with the file type 'Supporting Information'. Please ensure that each Supporting Information file has a legend listed in the manuscript after the references list.

6) Figure 1: Please confirm (a) that you are the photographer; or (b) provide written permission from the photographer to publish the photo(s) under our CC BY 4.0 license.

**Reviewers' Comments:**

Reviewer's Responses to Questions

**Key Review Criteria Required for Acceptance?**

**Methods**

-Are the objectives of the study clearly articulated with a clear testable hypothesis stated?

-Is the study design appropriate to address the stated objectives?

-Is the population clearly described and appropriate for the hypothesis being tested?

-Is the sample size sufficient to ensure adequate power to address the hypothesis being tested?

-Were correct statistical analysis used to support conclusions?

-Are there concerns about ethical or regulatory requirements being met?

Reviewer #1: Line 115. Can probably just omit the clause before the comma and start with We conducted

Line 127. Because half of the expert-suggested references did not appear in the literature search, this begs the question – how many other relevant references were missed with the search terms/databases used? How did the authors verify that their search was comprehensive?

Line 153. While I agree that it doesn’t make sense to pool results from different trap types (eg EWT and tent traps) because they work through inherently different mechanisms, I think it would make sense to pool results from similar traps and attempt a meta-analysis with a random effects model to account for variability in study and trap design (eg distance between traps/HLC, trap colour, trap size, odour bait, etc). Based on the outcomes reported, it would seem feasible to calculate standardised outcome metrics (ie IRR). Since EWT had the most comparisons, this could be attempted for this trap type, and a narrative synthesis could then be used for the others. Did the authors attempt this, and what was the result?

Reviewer #2: The manuscript offers a systematic review of all published studies comparing alternative trapping methods for molecular xenomonitoring. Objectives are clearly identified, and the methods are appropriate to achieve the goals of the study. Although the authors originally attempted to perform a meta-analysis across all published studies, this was not possible due to data scarcity and heterogeneity of experimental designs across studies.

Reviewer #3: The objectives are clearly stated

The study design is appropriate for the objective

The selection process for articles is clearly described

The methods for synthesizing the findings support the conclusions reached

No ethical concerns

**Results**

-Does the analysis presented match the analysis plan?

-Are the results clearly and completely presented?

-Are the figures (Tables, Images) of sufficient quality for clarity?

Reviewer #1: line 179. What is the definition of a “comparison”?

line 187. For those where the habitat was unspecified, did the papers provide GPS coordinates? This may enable inference about what habitat the collections were performed in.

line 194. “At least” or “over”?

line 224. For study 13, what is meant by “mean monthly biting rate” for catches that didn’t involve HLC? For those studies where “total catch” was the metric, did the authors verify that these comparisons were done 1:1? For this table, it would be useful to have more characteristics of the included studies listed. For example, which trap types the study included

Line 228. A 1:1 ratio of sampling effort or traps? If sampling effort, this would mean the traps/HLC were each performed for the same number of days and in the same quantities each day.

Line 245. Were there any qualities of the EWT studies that were more prevalent in the comparisons that demonstrated no sig diff/more effective? Eg did most of these include CO2, etc. May be difficult to say, but thought it might be good to tie back in Table 5. Similar question on the other studies too – did trap characteristics impact whether the trap was non inferior to HLC? For example, the Atekem study showed that black trousers were statistically better than blue or blue/black, but this isn’t mentioned. This comment also relates to an earlier comment about the possibility of implementing a random effects meta-analysis, at least on EWT studies.

Reviewer #2: The analysis carried out is adequate considering the data available across the literature. Although the authors intended to perform a meta-analysis comparing the performance of alternative catching methods vs. HLC in different studies, this was not possible due to the scarcity and heterogeneity of data.

My only suggestion would be linking specific heterogeneity findings to individual studies (using the Ref.). This would greatly enhance the manuscript's transparency and utility. It would allow readers to understand not just that there's variation across studies, but also to identify which specific studies contribute to that variation and potentially understand why (based on their methodologies, locations, species, etc.). This level of detail would transform this review from a general summary into a more actionable reference tool for future research design and intervention planning (see below).

192-194. It might be useful to indicate references for the different experimental designs. For example, when the distance between the comparing trapping methods is as small as 2m, the statistical test should consider that the events are dependent; i.e., each fly is provided with a choice to land on "A" OR "B". Conversely, if the sampling methods are sufficiently far apart (e.g., 50m), catches on "A" should not affect "B". Different statistical methods should be used depending on the relative location of the sampling methods.

194-195: Would it be useful to provide the references for the six studies in which the distance between the trapping methods was not specified? This information seems important for selecting the appropriate statistical test.

200-202: As above.

204-207: As above.

227-230: References to the particular studies would be useful.

TABLES

Tables 3-5. Add references to the studies. See Table 6 as an example.

Reviewer #3: The analysis is comprehensive

Yes, the results are clearly presented

Need to work on the design or structure of the tables, the figures are good by visually judging

**Conclusions**

-Are the conclusions supported by the data presented?

-Are the limitations of analysis clearly described?

-Do the authors discuss how these data can be helpful to advance our understanding of the topic under study?

-Is public health relevance addressed?

Reviewer #1: The authors accurately discuss the implications of the results and do not overstate or otherwise misconstrue their findings. The authors discuss how the findings are useful in directing further research.

Reviewer #2: The discussion and conclusion sections provide an accurate summary of the findings. Perhaps more importantly than stating the "known", they also highlight the "unknown" and provide recommendations for future studies, such as the need for standardising methodologies.

Reviewer #3: Conclusions are supported by the data

The limitations are very clearly described

The discussion clearly laid out the current situation with Blackfly collection and highlights how their research is advancing the discourse

The findings and discussion are critical for advancing tools for assessing progress in onchocerciasis elimination

**Editorial and Data Presentation Modifications?**

Reviewer #1: line 69. Xenomonitoring (“foreign” monitoring) refers to the detection of pathogen DNA in haematophagous insects as a proxy for human infection. What is done in oncho programs is technically not xenomonitoring, because the goal is to estimate transmission rather than human infection. If programs had guidance around how to use whole body (rather than just head) diagnostics, then this would be xenomonitoring. Small detail, but I think important to clarify this in the manuscript.

Line 78. Reference for the sentence that ends in biting density?

Reviewer #2: INTRODUCTION

60 & 62. (Just semantics…) Although the microfilaricidal effect of ivermectin results in a reduction of transmission, I wouldn't say that it is a "preventive" drug if we consider the words "preventive" and "prophylactic" synonymous in this context. I know this is the term used in reference [1], but I'm still not convinced. Although it "prevents" transmission, it does not prevent new infections.

79. Similar to above, as far as I know, the prophylactic effect of ivermectin (or doxycycline, or any other drug) has not been clearly demonstrated.

73-84. If I may, I would suggest including pruritus as an adverse effect of the HLC method. As a former collector, I can confirm that the itching afterwards is definitely unpleasant. Note that "blackfly fever" is described as a condition following high exposure to blackfly saliva. Authors can find references in the literature.

Figure 1. Verify that index letters (a, b, c...) correctly correspond to the designated photos. I.e., there are two "c" and no "h".

RESULTS

Mexico is actually part of North America (not South America). Same for Table 3.

Reviewer #3: The paper is well written, I recommend it for publication.

**Summary and General Comments**

Reviewer #1: This manuscript presents a systematic review of the evidence on the effectiveness of different trapping technologies for Simulium blackflies. The authors objectively weigh the evidence available and the limitations of the existing studies, and present their findings clearly while not overstating the significance of their results. They highlight how their findings can be used to inform further research in this field. This manuscript is a valuable addition to the literature.

Reviewer #2: This systematic review is very timely. Stakeholders involved in the elimination of onchocerciasis, mostly in Africa, are trying to align their goals and activities with the roadmap towards elimination, depending on their epidemiological situation—that is to say, whether they are in the accelerating or validating interruption of transmission phases. Molecular xenomonitoring (MX) has been proposed as a key tool to assess epidemiological status. However, to effectively assess transmission dynamics using entomological indicators, the challenges begin with the need to collect large numbers of vectors, i.e., ~6,000 per transmission zone. This issue can be subdivided into at least three key components: (i) requirement for effective catching methods that can easily harvest the required number of blackflies; (ii) methods that selectively collect anthropophilic blackflies likely to be vectors of human disease, comparable to catches obtained with HLC; and (iii) methods that are cost-effective and easy to implement with limited external support.

HLC scores highly on the three aforementioned criteria and therefore has been the gold standard trapping method since the beginning of the OCP. However, ethical concerns associated with exposing collectors to bites have motivated the search for alternatives. This manuscript provides the first -- to my knowledge -- systematic review of alternative catching methods in the last ~15 years. There have been other reviews, but they were more descriptive rather than comparative. The manuscript provides several lessons:

- Despite widespread interest in identifying alternative catch methods, only 13 articles were identified, describing 79 individual comparisons across eight countries in Africa and the Americas. Although there are some older behavioural studies that haven't been included in the review (e.g., Thompson et al.), I believe the scope of this article focuses on alternative trapping methods rather than other behavioural characteristics, such as host-seeking behaviour. Therefore, the selection of articles seems appropriate.

- The high heterogeneity in research methods (rotation strategies, distance between treatments…) and sometimes lack of crucial data (e.g., season, species, habitat, trap design, use of lures, statistical tests…) make it extremely difficult to compare different methods across the available literature. This heterogeneity prevented the possibility of a formal meta-analysis, as mentioned.

- There is also high heterogeneity in the outputs used as indicators (e.g., total catch, mean hourly catch, mean daily catch, or monthly biting rate). Therefore, the combined disparity in research methods, indicators, and sometimes the lack of information made it impossible for the authors to perform a formal meta-analysis (as discussed in the article).

- Alternative methods performed generally more poorly than HLC. However, performance was inconsistent across studies, and some comparative studies suggest that combinations of several alternative traps could outperform HLC (at least in some circumstances?).

- Early studies (e.g., B.H. Thompson et al. in the 1970s-1980s) suggested that trap method performance is mediated by host-seeking behaviour. Therefore, different cytospecies, host preferences, habitat, season, etc. may condition how well the technology works (e.g., according to Thompson, forest cytospecies of S. damnosum s.l. seemed to respond better to odours than savannah cytospecies). Unfortunately, there are insufficient data in the literature to assess whether alternative methods may perform better under certain circumstances (e.g., for certain species, habitats, or seasons). This manuscript helps us realise this knowledge gap.

- Ideally, the trapping method should selectively collect human vectors. The HLC method is biased towards anthropophilic flies, making it suitable for using MX as a proxy for O. volvulus transmission dynamics. Unfortunately, there is a lack of comparative data to determine whether alternative methods maintain this selective sampling advantage or inadvertently collect a broader range of blackfly species with different feeding preferences (i.e., whether the fraction of flies collected by alternative methods was comparable to that of HLC).

Therefore, this manuscript not only provides what we know about alternative catching methods, but also what we don't know. It also highlights the importance of standardising study methods in order to allow comparisons. The authors accomplish this with easily readable, well-written, and concise text without unnecessary content.

In addition, and as the authors suggest, further studies will be required to assess the logistical implications and financial feasibility of alternative strategies.

For all the above reasons, I believe making this ms. public will enormously help all stakeholders (mostly national control programmes) design their interventions and monitoring strategies.

My main suggestion for amendment would be adding references to particular studies when discussing differences. This is a very minor correction and shouldn't take too long to implement. More details above.

Reviewer #3: The problem is clearly stated, the methodology and its weaknesses (and why) are clearly described, the findings, the implications for onchocerciasis elimination validation are clearly laid out. This work will contribute significantly to what we know about evaluating the traps, consequently, it highlighted the need for standardization in the processes and reporting of studies on evaluation of tools for Blackfly sampling, to support meta-analysis of the body of evidence on effectiveness of these traps.

PLOS authors have the option to publish the peer review history of their article (what does this mean?). If published, this will include your full peer review and any attached files.

Reviewer #1: No

Reviewer #2: **Yes:** Inaki Tirados

Reviewer #3: No

**Figure resubmission:** While revising your submission, we strongly recommend that you use PLOS’s NAAS tool (https://ngplosjournals.pagemajik.ai/artanalysis) to test your figure files. NAAS can convert your figure files to the TIFF file type and meet basic requirements (such as print size, resolution), or provide you with a report on issues that do not meet our requirements and that NAAS cannot fix.
---

## [Decision Letter · Decision Letter 1]

11 May 2026

Dear Dr. Hawkes,

We are pleased to inform you that your manuscript 'A systematic review comparing the performance of alternative blackfly (*Simulium*) trapping methods against the standard human landing catch (HLC) for onchocerciasis surveillance' has been provisionally accepted for publication in PLOS Neglected Tropical Diseases.

Best regards,

Emeka John Dingwoke, Ph.D

Academic Editor

Amy Morrison

Section Editor

Shaden Kamhawi

co-Editor-in-Chief

Paul Brindley

co-Editor-in-Chief

Reviewer's Responses to Questions

**Key Review Criteria Required for Acceptance?**

**Methods**

-Are the objectives of the study clearly articulated with a clear testable hypothesis stated?

-Is the study design appropriate to address the stated objectives?

-Is the population clearly described and appropriate for the hypothesis being tested?

-Is the sample size sufficient to ensure adequate power to address the hypothesis being tested?

-Were correct statistical analysis used to support conclusions?

-Are there concerns about ethical or regulatory requirements being met?

Reviewer #1: (No Response)

Reviewer #2: Objectives are clearly stated. Methods are justified and fitting for the study. The search methods were appropriate to identify relevant studies.

**Results**

-Does the analysis presented match the analysis plan?

-Are the results clearly and completely presented?

-Are the figures (Tables, Images) of sufficient quality for clarity?

Reviewer #1: (No Response)

Reviewer #2: To my knowledge, all relevant studies were identified. The results section appropriately outlines the objectives of this review. References were added in this section, improving the clarity of the article.

**Conclusions**

-Are the conclusions supported by the data presented?

-Are the limitations of analysis clearly described?

-Do the authors discuss how these data can be helpful to advance our understanding of the topic under study?

-Is public health relevance addressed?

Reviewer #1: (No Response)

Reviewer #2: In the discussion and conclusion sections, the authors correctly summarise the gaps in knowledge concerning trapping methods and the implications for the roadmap towards elimination. The conclusions from this review will help national programmes and other stakeholders to plan interventions, but also field-based operational research that may help to fill the gaps and improve the sustainability of xenomonitoring, especially in near- or post-elimination settings.

**Editorial and Data Presentation Modifications?**

Reviewer #1: (No Response)

Reviewer #2: n/a

**Summary and General Comments**

Reviewer #1: (No Response)

Reviewer #2: Although this systematic review may seem to include only a few studies, to my knowledge they represent all the relevant papers that are publicly available. Previously, a couple of articles have discussed different trapping methods, which were useful in outlining some technical aspects of alternative catching strategies. However, this manuscript provides the first systematic review on the topic, offering an objective assessment of the efficacy of alternative methods compared to the gold standard, HLC. More importantly, this work identifies knowledge gaps and delivers useful recommendations for national programmes, research institutions and other stakeholders that will help consolidate the roadmap towards elimination. The amendments in this version, especially the addition of references to particular results, add clarity to the article.

PLOS authors have the option to publish the peer review history of their article (what does this mean?). If published, this will include your full peer review and any attached files.

Reviewer #1: No

Reviewer #2: **Yes:** Inaki Tirados

---

## [Editor Report · Acceptance letter]

Dear Dr. Hawkes,

We are delighted to inform you that your manuscript, "A systematic review comparing the performance of alternative blackfly (*Simulium*) trapping methods against the standard human landing catch (HLC) for onchocerciasis surveillance," has been formally accepted for publication in PLOS Neglected Tropical Diseases.

Best regards,

Shaden Kamhawi

co-Editor-in-Chief

Paul Brindley

co-Editor-in-Chief
